# Zero-Shot Cyclic Peptide Design via Composable Geometric Constraints

**Dapeng Jiang** [1 2 *] **Xiangzhe Kong** [1 3 *] **Jiaqi Han** [4 *] **Mingyu Li** [1] **Rui Jiao** [1 3] **Wenbing Huang** [5 6]
**Stefano Ermon** [4] **Jianzhu Ma** [1 7] **Yang Liu** [1 3]

## Abstract

Cyclic peptides, characterized by geometric constraints absent in linear peptides, offer enhanced biochemical properties, presenting new opportunities to address unmet medical needs. However, designing target-specific cyclic peptides remains underexplored due to limited training data. To bridge the gap, we propose CP-Composer, a novel generative framework that enables zero-shot cyclic peptide generation via composable geometric constraints. Our approach decomposes complex cyclization patterns into unit constraints, which are incorporated into a diffusion model through geometric conditioning on nodes and edges. During training, the model learns from unit constraints and their random combinations in linear peptides, while at inference, novel constraint combinations required for cyclization are imposed as input. Experiments show that our model, despite trained with linear peptides, is capable of generating diverse target-binding cyclic peptides, reaching success rates from 38% to 84% on different cyclization strategies.

## 1. Introduction

Peptides occupy an intermediate position between small molecules and antibodies, offering unique advantages over conventional drug formats, such as higher specificity and enhanced cell permeability (Fosgerau & Hoffmann, 2015; Lee et al., 2019). Among them, cyclic peptides, which introduce geometric constraints into linear peptides, have earned

---
[*]Equal contribution [1]Institute for AI Industry Research (AIR), Tsinghua [2]Xingjian College, Tsinghua University [3]Department of Computer Science and Technology, Tsinghua University [4]Stanford University [5]Gaoling School of Artificial Intelligence, Renmin University of China [6]Beijing Key Laboratory of Research on Large Models and Intelligent Governance [7]Department of Electronic Engineering, Tsinghua University. Correspondence to: Yang Liu <liuyang2011@tsinghua.edu.cn >, Jianzhu Ma <majianzhu@tsinghua.edu.cn>.

*Proceedings of the $42^{nd}$ International Conference on Machine Learning*, Vancouver, Canada. PMLR 267, 2025. Copyright 2025 by the author(s).

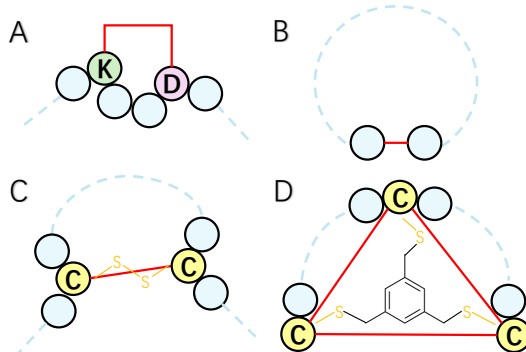

*Figure 1.* Four common strategies to form cyclic peptides. **(A)** Stapled peptide where a lysine (K) at position $i$ and an aspartic acid (D) at position $i+3$ are connected via dehydration condensation on side chains. The aspartic acid can also be replaced with glutamic acid (E) at position $i+4$. **(B)** Head-to-tail peptide where the first residue and the last residue form an amide bond for connection. **(C)** Disulfide peptide where two cysteines (C) non-adjacent in sequence are spatially connected through a disulfur bond. **(D)** Bicycle peptide which uses 1,3,5-trimethylbenezene to form a triangle between three cysteines (C) non-adjacent in sequence.

increasing attention (Zorzi et al., 2017). These constraints stabilize the peptide conformation, enhancing biochemical properties including binding affinity, *in vivo* stability, and oral bioavailability (Ji et al., 2024), which are essential for identifying desired drug candidates (Zhang & Chen, 2022).

Existing literature on target-specific peptide generation primarily focus on linear peptides, utilizing autoregressive models (Li et al., 2024a), multi-modal flow matching (Li et al., 2024b; Lin et al., 2024), and geometric latent diffusion (Kong et al., 2024). However, these methods are not directly applicable to cyclic peptide design due to the scarcity of available data (Rettie et al., 2024). Other approaches either impose geometric constraints on linear peptides through post-filtering (Wang et al., 2024b), which typically results in low acceptance rates, or rely on hard-coded model design (Rettie et al., 2024), which lacks generalizability across different cyclization patterns. In contrast, we hypothesize that the complex geometric constraints of cyclic peptides can be decomposed into fundamental unit constraints, resembling how complex mathematical formulas are built from basic arithmetic operations. While existing datasets rarely contain peptides that satisfy intricate cyclic

constraints, they typically include abundant instances of single unit constraints and their random combinations, which serve as the building blocks for more complicated designs. Therefore, we reason that a framework could potentially be developed to learn these unit constraints from available linear peptide data, circumventing data limitations and enabling generalization to the diverse combined constraints required for cyclic peptide design.

In this paper, we present CP-Composer, a framework for zero-shot cyclic peptide generation, relying solely on available data for linear peptides. Our work is equipped with the following contributions. **1) Decomposing cyclization strategies into fundamental geometric constraints.** We identify four common chemical cyclization strategies (Figure 1) and formalize cyclic peptide design as a geometrically constrained generation problem. By analyzing cyclization patterns, we derive two fundamental unit constraints, type constraints and distance constraints, allowing description of diverse cyclization strategies to be specific combinations of these units. **2) Encoding constraints with geometric conditioning.** We incorporate unit constraints into a the denoising network of a diffusion model (Kong et al., 2024) using additional vectorized embeddings of types and distances on geometric graphs, which enables flexible conditioning on compositions of constraints required for cyclic peptide generation. **3) Enabling zero-shot cyclic peptide design.** We jointly train conditional and unconditional models on unit constraints and their random combinations found in linear peptide data. At inference, novel constraint combinations corresponding to desired cyclization strategies, which are unseen during training, are imposed as input conditions. The model is guided by the difference in score estimates between conditional and unconditional models, enabling zero-shot generalization to cyclic peptides. **4) Assessing generated cyclic peptides on comprehensive metrics.** Experiments demonstrate that our CP-Composer generates cyclic peptides with complex geometric constraints effectively, achieving high success rates from 38% to 84%, while maintaining realistic distributions on amino acid types and dihedral angles. Molecular dynamics further confirm that the generated cyclic peptides exhibit desired binding affinity while forming more stable binding conformation compared to the native linear peptide binders.

## 2. Related Work

**Geometric diffusion models.** Besides their success on applications like image (Rombach et al., 2021; Song et al., 2020; 2021a) and video (Ho et al., 2022) generation, diffusion models have become a preeminent tool in modeling the distribution of structured data in geometric domains. While early works have explored their applicability on tasks like molecule generation (Xu et al., 2022; 2023; Park & Shen,

2024), there have been growing interests in scaling these models to systems of larger scales, such as antibody (Luo et al., 2022), peptide (Kong et al., 2024), and protein (Yim et al., 2023; Watson et al., 2023; Anand & Achim, 2022) in general, or to those with complex dynamics, such as molecular dynamics simulation (Han et al., 2024b). Despite fruitful achievements, how to impose diverse geometric constraints stills remain under-explored for geometric diffusion models, which we aim to address in this work.

**Diffusion guidance.** Diffusion sampling can be flexibly controlled by progressively enforcing guidance through the reverse denoising process. Dhariwal & Nichol (2021) proposes classifier-guidance, which employs an additionally trained classifier to amplify the guidance signal. Classifier-free guidance (CFG) (Ho & Salimans, 2022) is a more widely adopted alternative that replaces the classifier with the difference of the conditional and unconditional score, which has been further generalized to the multi-constraint scenario by composing multiple scores in diffusion sampling (Liu et al., 2022; Huang et al., 2023). Diffusion guidance has also been explore for solving inverse problems on images (Song et al., 2024; Kawar et al., 2022; Song et al., 2021b), molecules (Bao et al., 2022), and PDEs (Jiang et al., 2024). Our approach instead extends CFG to compose geometric constraints with application to cyclic peptide design.

**Peptide design.** Target-specific peptide design initially relied on physical methods using statistical force fields and fragment libraries (Hosseinzadeh et al., 2021; Swanson et al., 2022). With the rise of equivariant neural networks (Satorras et al., 2021; Han et al., 2024a), geometric deep generative models have emerged. PepFlow (Li et al., 2024b) and PPFlow (Lin et al., 2024) use multi-modal flow matching, while PepGLAD (Kong et al., 2024) applies geometric latent diffusion with a full-atom autoencoder. However, these methods struggle with cyclic peptide design due to limited data. Prior works introduce disulfide bonds via post-filtering (Wang et al., 2024b) or enforce head-to-tail cyclization through hard-codedo model design (Rettie et al., 2024). In contrast, our approach decomposes cyclization into fundamental unit constraints, enabling zero-shot cyclic peptide generation with broad flexibility across diverse patterns.

## 3. Method

In this section, we detail our method, CP-Composer. We first introduce basic concepts of peptide modeling and cyclic strategies in Sec. 3.1 and specify these strategies as constraints in Sec. 3.2. We further present the guided generation framework and the encoding strategy for incorporating the constraints in Sec. 3.3 and Sec. 3.4, respectively. We finally describe the training and inference schemes in Sec. 3.5. The overal workflow is depicted in Fig. 2.

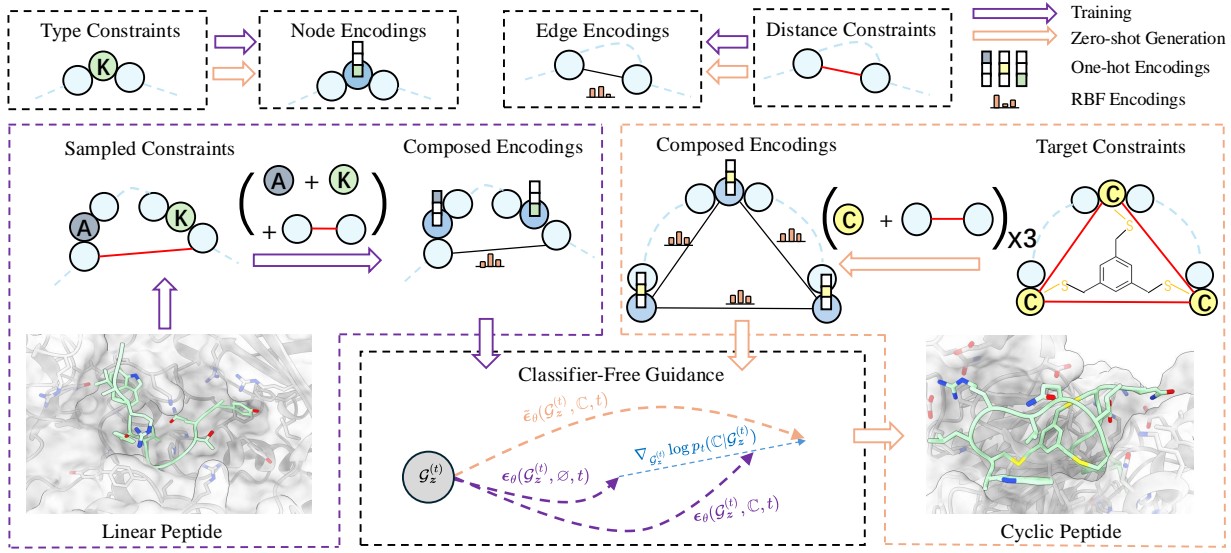

*Figure 2.* **Overall training and inference design of CP-Composer.** We define two unit constraints, type constraint and distance constraint (§ 3.2), which are incorporated into the diffusion model via geometric conditioning (§ 3.4). During training, the model learns from single unit constraints and their combinations observed in linear peptides. At inference, novel combinations corresponding to specific cyclization strategies are imposed with guidance signal amplified by classifier-free guidance, enabling zero-shot cyclic peptide design (§ 3.5).

## 3.1. Preliminaries

**Representing peptide as geometric graph.** We represent the binding site and peptide as a fully-connected geometric graph $\mathcal{G} = (\mathcal{V}, \mathcal{E})$ where $\mathcal{V}$ is the set of nodes and $\mathcal{E}$ is the set of edges. Each node is a residue, binded with node features $(\boldsymbol{h}_i, \vec{\boldsymbol{X}}_i)$ with $\boldsymbol{h}_i \in \mathbb{R}^m$ being the one-hot encoding of the amino acid type and $\vec{\boldsymbol{X}}_i \in \mathbb{R}^{k_i \times 3}$ being the coordinate of the $k_i$ atoms.

**Geometric latent diffusion model for peptide design.** Our model is built on PepGLAD (Kong et al., 2024), a latent geometric diffusion model, but is adaptable to other diffusion-based frameworks. It employs a variational autoencoder to project peptide graphs $\mathcal{G}$ into residue-level latents $\mathcal{G}_{\boldsymbol{z}} = \{(\boldsymbol{z}_i, \vec{\boldsymbol{z}}_i)\}_{i=1}^N$ with an encoder $\mathcal{E}_\phi$, and a corresponding decoder $\mathcal{D}_\xi$ for the inverse, where $\boldsymbol{z}_i \in \mathbb{R}^8$ is the E(3)-invariant latent and $\vec{\boldsymbol{z}}_i \in \mathbb{R}^3$ is the E(3)-equivariant counterpart. A diffusion model is learned in the compact latent space, with the denoiser $\boldsymbol{\epsilon}_\theta(\mathcal{G}_{\boldsymbol{z}}^{(t)}, t)$ parameterized by an equivariant GNN (Kong et al., 2023). The sampling process initiates with latents $\mathcal{G}_{\boldsymbol{z}}^{(T)} = \{(\boldsymbol{z}_i^{(T)}, \vec{\boldsymbol{z}}_i^{(T)})\}_{i=1}^N$ drawn from the prior and gradually denoises it using DDPM (Ho et al., 2020) sampler for a total of $T$ steps. The final latents $\mathcal{G}_{\boldsymbol{z}}^{(0)}$ are decoded back to the data space using decoder $\mathcal{D}_\xi$.

**Cyclic peptide and cyclization strategies.** Unlike common linear peptides, which are chain-like structures, a cyclic peptide is formed by animo acids connected in a ring structure. As shown in Fig. 1, we primarily focus on four types of cyclic peptides in this paper: stapled, head-to-tail, disulfide and bicycle peptides. Each strategy applies constraints on

specific amino acid types and/or their pairwise distances. Taking the *disulfide peptide* as an example (Fig. 1C), to link two cysteines at indices $i, j$ with a disulfur bond of length $d_S$, a disulfide peptide is constrained by[1]

$$\mathbb{C}_{\text{Disulfide},i,j} = (\{\arg\max(\boldsymbol{h}_i) = \arg\max(\boldsymbol{h}_j) = k_C\},$$
$$\{\|\vec{\boldsymbol{X}}_i - \vec{\boldsymbol{X}}_j\|_2 = d_S\}), \quad (1)$$

where $k_C$ represents the index of cysteine (C) in the one-hot embeddings. This constraint can be decomposed into two node-level constraints on the amino acid types and one edge-level constraint on the distance. We refer to these as *unit geometric constraints* with further details on these constraints provided in Sec. 3.2. We demonstrate that all four cyclic strategies can be expressed as combinations of these unit geometric constraints in Appendix B.

## 3.2. Decomposing Cyclization Strategies as Geometric Constraints

In this work, we consider two types of unit geometric constraints, namely *type constraint* and *distance constraint*. In particular, type constraint operates on node-level by enforcing the node to be of certain type, while distance constraint takes place on edge-level, specifying a pair of nodes to reside at a certain distance.

**Definition 3.1** (Type constraint). A type constraint is a set $\mathbb{C}_T := \{(i, l_i)\}_{i \in \mathcal{V}_T}$ where each entry $(i, l_i)$ represents that node $i$ should be of type $l_i$, while $\mathcal{V}_T \subseteq \mathcal{V}$ is the set of nodes to enforce the type constraint.

---

[1] For simplicity, we slightly abuse the notation $\|\vec{\boldsymbol{X}}_i - \vec{\boldsymbol{X}}_j\|_2$, where the distance is measured using the nodes' $C_\alpha$ coordinates.

**Definition 3.2** (Distance constraint). A distance constraint is a set $\mathbb{C}_D := \{(i, j, d_{ij})\}_{(i,j)\in\mathcal{E}_D}$ where each element $(i, j, d_{ij})$ represents that node $i$ and $j$ should be positioned at the distance of $d_{ij}$, while $\mathcal{E}_D \subseteq \mathcal{E}$ is the set of edges to enforce the distance constraint.

Notably, our taxonomy of geometric constraints is particularly interesting due to its *completeness*, in the sense that each of the cyclic strategies $\mathbb{C}$ described in Sec. 3.1 can be decomposed into combinations of type constraints $\mathbb{C}_T$ and/or distance constraints $\mathbb{C}_D$. We defer the detailed explanations to Appendix B.

**Problem definition.** We formulate the task of cyclic peptide design as finding the candidate peptides $\mathcal{G}$ that satisfy constraint $\mathbb{C}$, where $\mathbb{C}$ is any one of the four cyclic constraints.

### 3.3. Inverse Design with Diffusion Guidance

To perform inverse design, a widely adopted approach is to progressively inject certain guidance term into diffusion sampling towards the design target (Bao et al., 2022; Song et al., 2023), which share similar spirit as classifier guidance (Dhariwal & Nichol, 2021). Specifically, at each sampling step $t$, the conditional score is derived by Bayes' rule:

$$\nabla_{\mathcal{G}_{\mathbf{z}}^{(t)}} \log p_t(\mathcal{G}_{\mathbf{z}}^{(t)}|\mathbb{C}) = \nabla_{\mathcal{G}_{\mathbf{z}}^{(t)}} \log p_t(\mathcal{G}_{\mathbf{z}}^{(t)})$$
$$+ \nabla_{\mathcal{G}_{\mathbf{z}}^{(t)}} \log p_t(\mathbb{C}|\mathcal{G}_{\mathbf{z}}^{(t)}), \quad (2)$$

where the last term $\nabla_{\mathcal{G}_{\mathbf{z}}^{(t)}} \log p_t(\mathbb{C}|\mathcal{G}_{\mathbf{z}}^{(t)})$ takes the effect as guidance, which can typically be a hand-crafted energy function (Kawar et al., 2022; Song et al., 2024) or a pretrained neural network (Dhariwal & Nichol, 2021; Bao et al., 2022).

However, empirically the approach is often demonstrated unfavorable since the guidance term in Eq. 2 is the gradient of neural network, which detriments sample quality due to adversarial effect (Ho & Salimans, 2022). Distinct from the approach above, we propose an alternative that, inspired by classifier-free guidance, guides the sampling by directly composing unconditional and conditional score without additional gradient terms. In detail, we have,

$$\tilde{\boldsymbol{\epsilon}}_\theta(\mathcal{G}_{\mathbf{z}}^{(t)}, \mathbb{C}, t) = (w+1)\boldsymbol{\epsilon}_\theta(\mathcal{G}_{\mathbf{z}}^{(t)}, \mathbb{C}, t) - w\boldsymbol{\epsilon}_\theta(\mathcal{G}_{\mathbf{z}}^{(t)}, t) \quad (3)$$

where $w$ is the guidance weight and the guided score $\tilde{\boldsymbol{\epsilon}}_\theta$ will replace $\boldsymbol{\epsilon}_\theta$ for score computation. In particular, the rationale of Eq. 2 and Eq. 3 are linked by the following distribution

$$\tilde{p}_t(\mathcal{G}_z^{(t)}|\mathbb{C}) \propto p_t(\mathcal{G}_z^{(t)})p_t(\mathbb{C}|\mathcal{G}_z^{(t)})^w, \quad (4)$$

with the corresponding conditional score

$$\nabla_{\mathcal{G}_z^{(t)}} \log \tilde{p}_t(\mathcal{G}_z^{(t)}|\mathbb{C})$$
$$= \nabla_{\mathcal{G}_z^{(t)}} \log p_t(\mathcal{G}_z^{(t)}) + w\nabla_{\mathcal{G}_z^{(t)}} \log p_t(\mathbb{C}|\mathcal{G}_z^{(t)}),$$
$$\approx \boldsymbol{\epsilon}_\theta(\mathcal{G}_z^{(t)}, t) + w\nabla_{\mathcal{G}_z^{(t)}} \log p_t(\mathbb{C}|\mathcal{G}_z^{(t)}). \quad (5)$$

By further leveraging the relation $\nabla_{\mathcal{G}_z^{(t)}} \log p_t(\mathbb{C}|\mathcal{G}_z^{(t)}) = \nabla_{\mathcal{G}_z^{(t)}} \log p_t(\mathcal{G}_z^{(t)}|\mathbb{C}) - \nabla_{\mathcal{G}_z^{(t)}} \log p_t(\mathcal{G}_z^{(t)}) \approx \boldsymbol{\epsilon}_\theta(\mathcal{G}_z^{(t)}, \mathbb{C}, t) - \boldsymbol{\epsilon}_\theta(\mathcal{G}_z^{(t)}, t)$ into Eq. 5, we obtain the expression in Eq. 3.

Conceptually, Eq. 2 adopts energy-guidance that directly models $\log p_t(\mathbb{C}|\mathcal{G}_z^{(t)})$ by an externally trained energy function. Eq. 3 instead follows the convention in classifier-free guidance by rewriting $\nabla_{\mathcal{G}_z^{(t)}} \log p_t(\mathbb{C}|\mathcal{G}_z^{(t)}) = \nabla_{\mathcal{G}_z^{(t)}} \log p_t(\mathcal{G}_z^{(t)}|\mathbb{C}) - \nabla_{\mathcal{G}_z^{(t)}} \log p_t(\mathcal{G}_z^{(t)}) \approx \boldsymbol{\epsilon}_\theta(\mathcal{G}_z^{(t)}, \mathbb{C}, t) - \boldsymbol{\epsilon}_\theta(\mathcal{G}_z^{(t)}, t)$, which gives Eq. 3 after simplification.

In recent studies, how to obtain the conditional score $\boldsymbol{\epsilon}_\theta(\mathcal{G}_z^{(t)}, \mathbb{C}, t)$ still remains unclear. Notably, $\mathbb{C}$ is a complicated geometric constraint, which is fundamentally different from a class label (Ho & Salimans, 2022) or a target value (Bao et al., 2022), where an embedding (*e.g.*, one-hot for class label) can be readily adopted as the control signal to feed into the denoiser. In the following section, we will introduce our approach to encode type and distance constraint.

### 3.4. Encoding Constraints via Geometric Conditioning

To encode the constraints as control signals, we propose geometric conditioning that embeds the type and distance constraints into the denoiser through vectorization.

**Conditioning type constraints.** For type constraint $\mathbb{C}_T = \{(i, l_i)\}_{i\in\mathcal{V}_T}$ where $l_i \in \{0, 1, \cdots, K-1\}$ is the desired node type for node $i$, we operate at node-level by augmenting the E(3)-invariant node feature $\boldsymbol{h}_i$ with an additional vector $\boldsymbol{l}_i \in \mathbb{R}^K$ which serves as the control signal. This corresponds to the encoding function $f_T(\mathbb{C}_T) = \{(i, \boldsymbol{l}_i)\}_{i\in\mathcal{V}_T}$ that lifts $l_i$ to the embedding space where

$$\boldsymbol{l}_i = \begin{cases} \text{One-hot}(l_i) & i \in \mathcal{V}_T, \\ \boldsymbol{0} & i \in \mathcal{V}\backslash\mathcal{V}_T. \end{cases} \quad (6)$$

Such design of the control signal is simple yet effective, since different type constraints will induce different signal $\boldsymbol{l}_i$, thus making the constraints distinguishable to the network. More importantly, for any type constraint, the conditional score $\boldsymbol{\epsilon}_\theta(\mathcal{G}_{\mathbf{z}}^{(t)}, \mathbb{C}, t)$ obtained by this means still enjoys E(3)-equivariance, since $\boldsymbol{l}_i$ is E(3)-invariant.

**Conditioning distance constraints.** For distance constraint $\mathbb{C}_D := \{(i, j, d_{ij})\}_{(i,j)\in\mathcal{E}_D}$ where $d_{ij}$ specifies the distance between node $i$ and $j$, we instead design the encoding function as $f_D(\mathbb{C}_D) = \{(i, j, \boldsymbol{d}_{ij})\}_{(i,j)\in\mathcal{E}_D}$, where the control signal $\boldsymbol{d}_{ij}$ is defined at edge-level:

$$\boldsymbol{d}_{ij} = \begin{cases} \text{RBF}(d_{ij}) & (i, j) \in \mathcal{E}_D, \\ \phi & (i, j) \in \mathcal{E}\backslash\mathcal{E}_D. \end{cases} \quad (7)$$

Here RBF$(\cdot)$ is the radial basis kernel that lifts the distance from a scalar to a high-dimensional vector (Schütt et al., 2018), and $\phi$ denotes that the edges not in the set $\mathcal{E}_D$ will not be featurized. The control signal $\boldsymbol{d}_{ij}$ is then viewed as a special type of edge feature, which will be further processed by an additional dyMEAN layer (Kong et al., 2023), whose input will be the subgraph $(\mathcal{V}, \mathcal{E}_D)$ with edge features $\{\boldsymbol{d}_{ij}\}_{(i,j)\in\mathcal{E}_D}$. More details are deferred to Appendix C.2. Akin to the analysis for type constraints, our way of encoding distance constraints also preserve the E(3)-equivariance of the conditional score, with proof in Appendix A.2.

Moreover, the encoding is also injective, as formally stated in Theorem 3.3. Such property is crucial for effective guidance since different constraints will be projected as different control signals, always making them distinguishable to the score network.

**Theorem 3.3** (Injective)**.** *Both $f_T$ and $f_D$ are injective. That is, $f(\mathbb{C}^1) = f(\mathbb{C}^2)$ if and only if $\mathbb{C}^1 = \mathbb{C}^2$, where $(f, \mathbb{C}^1, \mathbb{C}^2)$ can be $(f_T, \mathbb{C}_T^1, \mathbb{C}_T^2)$ or $(f_D, \mathbb{C}_D^1, \mathbb{C}_D^2)$. Furthermore, their product function $\tilde{f}(\mathbb{C}_T, \mathbb{C}_D) := (f_T(\mathbb{C}_T), f_D(\mathbb{C}_D))$ is also injective.*

**Composing type and distance constraints.** Our approach of encoding the type and distance constraints in node- and edge-level respectively also facilitates conveniently composing them together. In particular, we can easily devise $\epsilon_\theta(\mathcal{G}_{\boldsymbol{z}}^{(t)}, \mathbb{C}_T, \mathbb{C}_D, t)$ by simultaneous feeding the type and distance control signals in Eq. 6 and 7 into the score network, which corresponds to enforcing a compositional constraint $(\mathbb{C}_T, \mathbb{C}_D)$. This extension is critical since it enables us to enforce richer combinations of the constraints at inference time, even generalizing to those unseen during training. In this way, we are able to design cyclic peptides with training data that only consist of linear peptides due to the generalization capability of our approach.

### 3.5. Training and Inference

With the geometric conditioning technique to derive the conditional score, we are now ready to introduce the training and inference framework.

**Design space for constraints.** For a linear peptide $\mathcal{G}$ sampled from training set with features $\{(\boldsymbol{h}_i, \vec{\boldsymbol{X}}_i)\}_{i=1}^N$, we consider the following design space for type constraint:

$$\mathcal{C}_T(\mathcal{G}) = \{\mathbb{C}_T | \mathbb{C}_T = \{(i, \arg\max(\boldsymbol{h}_i)\}_{i \in \mathcal{V}_T}, |\mathcal{V}_T| \leq 4\}, \tag{8}$$

which include all of the type constraints that control the type of the node to be the same as that of node $i$ in $\mathcal{G}$ and the number of constraints to be fewer or equal to 4. For distance

---

**Algorithm 1** Training Procedure of CP-Composer

**Input:** Data distribution $\mathcal{D}$, mask probabilities for type and distance constraints $p_T, p_D$, encoder $\mathcal{E}_\phi$, score network $\epsilon_\theta$, diffusion scheduler $\texttt{Scheduler}(\cdot)$

1: **while** not converged **do**
2:     Sample $\mathcal{G} \sim \mathcal{D}, \mathbb{C}_T \sim \text{Unif}(\mathcal{C}_T(\mathcal{G}))$,
    and $\mathbb{C}_D \sim \text{Unif}(\mathcal{C}_D(\mathcal{G}))$         {c.f. Eq. 8-9}
3:     $\mathbb{C}_T \leftarrow \varnothing$ with probability $p_T$
4:     $\mathbb{C}_D \leftarrow \varnothing$ with probability $p_D$
5:     $(\epsilon, \mathcal{G}_{\boldsymbol{z}}^{(t)}, t) \leftarrow \texttt{Scheduler}(\mathcal{E}_\phi(\mathcal{G}))$
6:     Take gradient step on
       $\mathcal{L}(\theta) = \|\epsilon - \epsilon_\theta(\mathcal{G}_{\boldsymbol{z}}^{(t)}, \mathbb{C}_T, \mathbb{C}_D, t)\|_2^2$
7: **end while**

---

constraint, we select the following design space:

$$\mathcal{C}_D(\mathcal{G}) = \{\mathbb{C}_D | \mathbb{C}_D = \{(i, j, \|\vec{\boldsymbol{X}}_i - \vec{\boldsymbol{X}}_j\|_2)\}_{(i,j)\in\mathcal{E}_D},$$
$$d_{\mathcal{G}}(i,j) \in \{3, 4, 6\}, |\mathcal{E}_D| \leq 6\}, \tag{9}$$

which spans across all possible distance constraints that specify the distance between node $i$ and $j$ to be their Euclidean distance in $\mathcal{G}$, while the shortest path distance between $i$ and $j$, *i.e.*, $d_{\mathcal{G}}(i,j)$, equals to 3, 4, or 6. We design $\mathcal{C}_T(\mathcal{G})$ and $\mathcal{C}_D(\mathcal{G})$ such that $\mathcal{C}_T(\mathcal{G}) \times \mathcal{C}_D(\mathcal{G})$ covers the constraint space of cyclic peptides, where $\times$ is the Cartesian product. This permits our approach to generalize to novel compositions within the space $\mathcal{C}_T(\mathcal{G}) \times \mathcal{C}_D(\mathcal{G})$ at inference time without necessarily seeing such particular combination in training data, *e.g.*, the four compositional constraints of cyclic peptides.

**Training.** We employ a single network $\epsilon_\theta$ to jointly optimize the conditional and unconditional score during training, following the paradigm in Ho & Salimans (2022). At each training step, we first sample $\mathcal{G}$ from training data distribution $\mathcal{D}$ and derive the candidate constraints $\mathcal{C}_T(\mathcal{G})$ and $\mathcal{C}_D(\mathcal{G})$. We then sample a type constraint $\mathbb{C}_T$ and a distance constraint $\mathbb{C}_D$ uniformly from the candidates $\mathcal{C}_T(\mathcal{G})$ and $\mathcal{C}_D(\mathcal{G})$, respectively. To jointly optimize the conditional and unconditional score networks, we replace $\mathbb{C}_T$ and $\mathbb{C}_D$ by empty set $\varnothing$ with probability $p_T$ and $p_D$ respectively, where the empty set will enforce no meaningful type and/or distance control signal which degenerates to the unconditional score. Finally, we encode $\mathcal{G}$ into latent space by $\mathcal{E}_\phi$, sample the noise $\epsilon$ and diffusion step $t$, and compute the noised latent $\mathcal{G}_{\boldsymbol{z}}^{(t)}$. The noise prediction loss (Ho et al., 2020) is adopted to train the score network. We present the detailed training procedure in Alg. 1.

**Inference.** At inference time, we will select one of the four cyclic constraints at one time. Each constraint is represented by $(\mathbb{C}_T^*, \mathbb{C}_D^*)$ where $\mathbb{C}_T^*$ and $\mathbb{C}_D^*$ are the target type and distance constraint, respectively. We start from the initial latent $\mathcal{G}_{\boldsymbol{z}}^{(T)}$ sampled from the prior and perform standard

*Table 1.* Success rates and KL divergence for generated samples from different cyclization strategies.

| | Stapled peptide | | | | Head-to-tail peptide | | | |
|---|---|---|---|---|---|---|---|---|
| | Succ. | AA-KL | B-KL | S-KL | Succ. | AA-KL | B-KL | S-KL |
| PepGLAD (Kong et al., 2024) | 22.80% | 0.1035 | 1.1401 | 0.0126 | 30.23% | 0.1052 | 1.1347 | 0.0125 |
| w/ EG (Bao et al., 2022) | 25.41% | 0.0744 | 1.1821 | 0.0127 | 61.63% | 0.0798 | 1.0891 | 0.0128 |
| CP-Composer $w = 0.0$ | 25.71% | 0.0932 | 1.1179 | 0.0126 | 37.21% | 0.1021 | 1.0787 | 0.0118 |
| CP-Composer $w = 1.0$ | 30.00% | 0.1017 | 1.1235 | 0.0161 | 55.81% | 0.1008 | 1.0604 | 0.0124 |
| CP-Composer $w = 2.0$ | 21.42% | 0.1067 | 1.0996 | 0.0147 | 65.11% | 0.1055 | 1.1005 | 0.0126 |
| +CADS (Sadat et al., 2024) | 27.14% | 0.0807 | 1.0975 | 0.0119 | 45.54% | 0.0798 | 1.0589 | 0.0132 |
| CP-Composer $w = 5.0$ | **38.57%** | 0.1812 | 1.1515 | 0.0180 | **74.42%** | 0.1320 | 1.0523 | 0.0122 |
| CP-Composer $w = 10.0$ | 32.86% | 0.3532 | 1.1726 | 0.0232 | 68.60% | 0.1784 | 1.0301 | 0.0175 |
| | Disulfide peptide | | | | Bicycle peptide | | | |
| | Succ. | AA-KL | B-KL | S-KL | Succ. | AA-KL | B-KL | S-KL |
| PepGLAD (Kong et al., 2024) | 0 | 0.0808 | 1.1324 | 0.0124 | 0 | 0.0838 | 1.1823 | 0.0238 |
| w/ EG (Bao et al., 2022) | 0 | 0.0711 | 1.0891 | 0.0103 | 0 | 0.0729 | 1.0968 | 0.0228 |
| CP-Composer $w = 0.0$ | 7.50% | 0.1016 | 1.1062 | 0.0151 | 0 | 0.1225 | 1.1980 | 0.0252 |
| CP-Composer $w = 1.0$ | 21.25% | 0.1477 | 1.0939 | 0.0151 | 11.53% | 0.1638 | 1.1490 | 0.0395 |
| CP-Composer $w = 2.0$ | 41.25% | 0.2873 | 1.0994 | 0.0379 | 30.76% | 0.2147 | 1.1195 | 0.0735 |
| +CADS (Sadat et al., 2024) | 3.75% | 0.0939 | 1.0788 | 0.0162 | 3.85% | 0.0901 | 1.0624 | 0.0684 |
| CP-Composer $w = 5.0$ | **82.50%** | 0.5139 | 1.0397 | 0.1913 | **84.62%** | 0.3385 | 1.0759 | 0.3351 |
| CP-Composer $w = 10.0$ | 62.50% | 1.6965 | 4.0312 | 1.1046 | 38.46% | 1.2677 | 8.1935 | 0.3374 |

---

**Algorithm 2** Inference Procedure of CP-Composer

**Input:** Target type and distance constraint $(\mathbb{C}_T^*, \mathbb{C}_D^*)$, diffusion sampler $\texttt{Sampler}(\cdot)$, guidance weight $w$, step $T$, score network $\boldsymbol{\epsilon}_\theta$, decoder $\mathcal{D}_\xi$

1: Initialize latents $\mathcal{G}_{\boldsymbol{z}}^{(T)}$ from prior
2: **for** $t = T, T-1, \cdots, 1$ **do**
3:     Compute score $\tilde{\boldsymbol{\epsilon}} \leftarrow (w+1)\boldsymbol{\epsilon}_\theta(\mathcal{G}_{\boldsymbol{z}}^{(t)}, \mathbb{C}_T^*, \mathbb{C}_D^*, t) - w\boldsymbol{\epsilon}_\theta(\mathcal{G}_{\boldsymbol{z}}^{(t)}, \varnothing, \varnothing, t)$ {Eq. 10}
4:     $\mathcal{G}_{\boldsymbol{z}}^{(t-1)} \leftarrow \texttt{Sampler}(\mathcal{G}_{\boldsymbol{z}}^{(t)}, \tilde{\boldsymbol{\epsilon}}, t)$    {Denoising step}
5: **end for**
**Return:** $\mathcal{D}_\xi(\mathcal{G}_{\boldsymbol{z}}^{(0)})$

---

diffusion sampling with the guided score:

$$\tilde{\boldsymbol{\epsilon}}(\mathcal{G}_{\boldsymbol{z}}^{(t)}, \mathbb{C}_T^*, \mathbb{C}_D^*, t) = (w+1)\boldsymbol{\epsilon}_\theta(\mathcal{G}_{\boldsymbol{z}}^{(t)}, \mathbb{C}_T^*, \mathbb{C}_D^*, t) - w\boldsymbol{\epsilon}_\theta(\mathcal{G}_{\boldsymbol{z}}^{(t)}, \varnothing, \varnothing, t), \quad (10)$$

where a modified classifier-free guidance is employed to further amplify the guidance signal. The sample is acquired by decoding $\mathcal{G}_{\boldsymbol{z}}^{(0)}$ back to the data space using the decoder $\mathcal{D}_\xi$. The inference procedure is depicted in Alg. 2.

## 4. Experiments

**Task.** We evaluate CP-Composer on target-specific cyclic peptide design, aiming to co-design the sequence and the binding structure of cyclic peptides given the binding site on the target protein.

**Dataset.** We utilize PepBench and ProtFrag datasets (Kong et al., 2024) for training and validation, with the LNR dataset (Kong et al., 2024; Tsaban et al., 2022) for testing. **PepBench** contains 4,157 protein-peptide complexes for training and 114 complexes for validation, with a target protein longer than 30 residues and a peptide binder between 4 to 25 residues. **ProtFrag** encompasses 70,498 synthetic samples resembling protein-peptide complexes, which are extracted from local contexts in protein monomers. **LNR** consists of 93 protein-peptide complexes curated by domain experts, with peptide lengths ranging from 4 to 25 residues.

We evaluate zero-shot cyclic peptide generation in Sec. 4.1, demonstrate the flexibility of composable geometric constraints with high-order multi-cycle constraints in Sec. 4.2, and assess the stability and binding affinity of the generated cyclic peptides through molecular dynamics in Sec. 4.3.

### 4.1. Zero-Shot Cyclic Peptide Generation

**Metrics.** We evaluate the generated peptides based on two key aspects: cyclic constraint satisfaction and generation quality. For each target protein in the test set, we generate five candidate peptides and compute the following metrics. **Success Rate (Succ.)** measures the proportion of target proteins for which at least one of the five generated peptides satisfies the geometric constraints of the specified cyclization strategy. **Amino Acid Divergence (AA-KL)** calculates the Kullback–Leibler (KL) divergence between the amino acid composition of reference peptides and all of the generated samples. For cyclization patterns that impose amino acid constraints at specific positions, we exclude these constrained amino acid types when computing the distributions, as successful designs inherently deviate from the reference distribution on these amino acid types. **Backbone Dihe-**

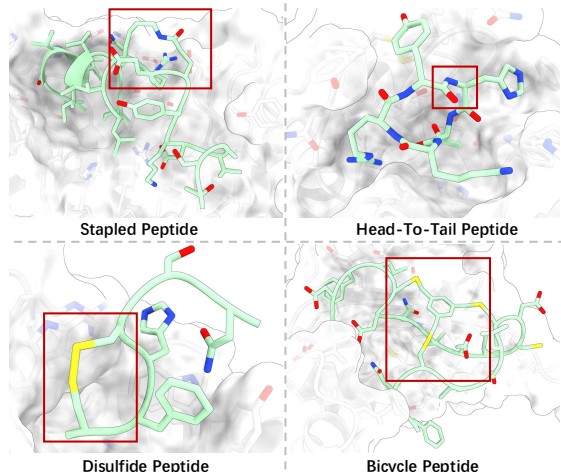

*Figure 3.* Four types of generated cyclic peptides, with the red boxes highlighting the position for cyclization.

**dral Angle Divergence (B-KL)** and **Side-Chain Dihedral Angle Divergence (S-KL)** indicate the KL divergence between the distribution of the dihedral angles in reference peptides and the generated samples, assessing rationality in the generated backbone and side chains, respectively.

**Baselines.** First, we compare our CP-Composer with the backbone model PepGLAD (Kong et al., 2024) without additional guidance to validate the effectiveness of our framework with composable geometric constraints. We further implement a baseline with the prevailing Energy-based Guidance (EG) (Dhariwal & Nichol, 2021; Bao et al., 2022) applied to node embeddings and pairwise distances to assess the advantages of our approach, with implementation details in Appendix C. To compare CP-Composer with other cyclic peptide generation method, we implement DiffPep-Builder (Wang et al., 2024a), a model specifically designed for disulfide peptides. Furthermore, we also incorperate our method with a advanced sampler Condition Annealed Diffusion Sampler(CADS) (Sadat et al., 2024) to analysis the performance of our method combining with other sampler.

**Results.** As shown in Table 1, CP-Composer significantly improves constraint satisfaction rates across all cyclization strategies compared to unguided baselines, while maintaining fidelity to reference distributions in amino acid composition and structural dihedral angles. The energy-guided baseline proves effective in simple cases requiring control over a single pairwise distance (i.e., head-to-tail cyclization), but struggles with more complex scenarios involving combinations of distance constraints and type constraints. This limitation is evident from its lower success rates on stapled peptides and complete failure in handling more intricate cyclization patterns including disulfide and bicycle peptides. In contrast, CP-Composer consistently achieves high success rates across these challenging cases, demonstrating the strength of our framework design with compos-

able geometric constraints. In Table 3, we further compare CP-Composer with DiffPepBuilder (Wang et al., 2024a). Although DiffPepbuilder is a method specifically designed for disulfide peptide generation, CP-Composer shows a better success rates than DiffPepbuilder. These results show the effectiveness of CP-Composer. We visualize examples of generated peptides for each cyclization strategy in Fig. 3, with more cases in Appendix E. Furthermore, the weight parameter $w$ effectively balances success rates and generation quality, with increasing control strength yielding higher constraint satisfaction yet slightly higher KL divergence, indicating a trade-off between constraint satisfaction and distributional fidelity. This flexibility allows users to customize the method based on specific application needs, prioritizing either higher success rates or closer resemblance to natural peptide distributions.

### 4.2. Flexibility in High-Order Combinations

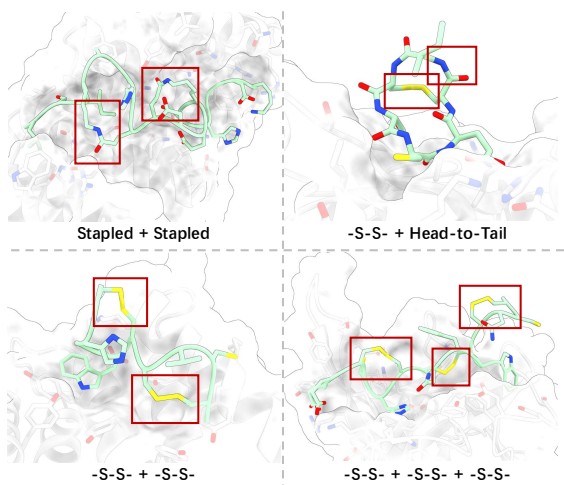

*Figure 4.* Generated peptides conforming to high-order combinations of cyclizations, with the red boxes highlighting the positions for cyclization.

**Setup.** To demonstrate the flexibility of our framework in handling composable geometric constraints, we investigate more complex and customized scenarios that involve multiple cyclizations within a single peptide. Specifically, we explore the following high-order combinations: **2\*Stapled** has two stapled pairs in one peptide. **-S-S- + H-T** includes one disulfide bond and one head-to-tail in one peptide; **2\*-S-S-** contains two disulfide bonds in one peptide; **3\*-S-S-** involves three disulfide bonds in one peptide; The flexibility of CP-Composer enables seamless implementation of these complex constraints: simply combining the individual unit constraints for each cyclization strategy allows the model to accommodate them simultaneously.

**Results.** As shown in Table 2, despite the increasing complexity of the constraints, CP-Composer achieves reasonable

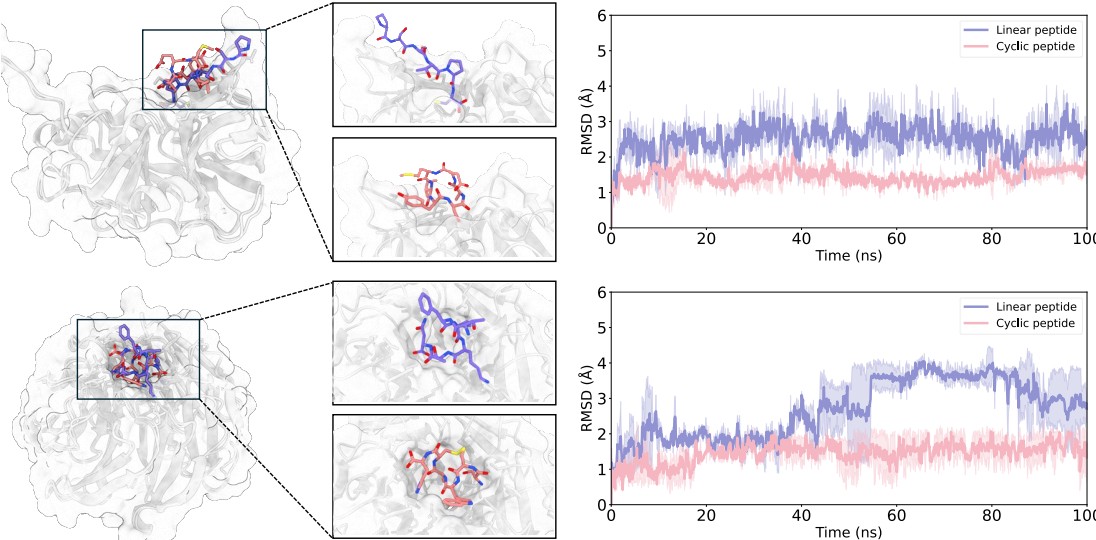

*Figure 5.* RMSD trajectories from 100 ns molecular dynamics simulations for two target proteins, each bound to either a native linear peptide binder or a cyclic peptide generated by our model. The target proteins and their corresponding linear peptide binders are derived from PDB 3RC4 (top) and PDB 4J86 (bottom), respectively.

success rates across all high-order cyclization scenarios. The control strength parameter $w$ remains effective, with higher values leading to enhanced success rates. The only exception is 2*Stapled, likely due to the inherent difficulty of the Staple strategy, which already exhibits the lowest success rate in Table 1. This indicates that our framework effectively learns to generate peptides that conform to the joint distribution of multiple constraints. Fig. 4 visualizes peptides with these high-order cyclization patterns, highlighting the flexibility of CP-Composer in designing structurally feasible peptides tailored for customized requirements.

*Table 2.* Success rates for high-order combinations of multiple cyclizations within the same peptide.

|          | 2*Stapled | -S-S-+H-T | 2*-S-S- | 3*-S-S- |
|----------|-----------|-----------|---------|---------|
| $w = 1.0$ | 2.5%      | 0         | 0       | 0       |
| $w = 2.0$ | **7.5%**  | 10.0%     | 26.0%   | 17.2%   |
| $w = 2.5$ | **7.5%**  | 20.0%     | 34.0%   | 34.5%   |
| $w = 3.0$ | **7.5%**  | **26.0%** | **62.0%** | **65.5%** |

*Table 3.* Success rates comparison between DiffPepBuilder and our method

| Succ. | Disulfide Peptide | 2*-S-S- |
|-------|-------------------|---------|
| CP-Composer | 41.25% | 62.00% |
| DiffPepBuilder (Wang et al., 2024a) | 23.07% | 32.78% |

In Table 3, we compare CP-Composer with DiffPepBuilder. The results show that our method outperform the cycplic peptide generation model under high-order cyclization scenario: two disulfide bonds in one peptide. This indicates the flexibility of our framework.

### 4.3. Evaluations by Molecular Dynamics

**Setup.** We perform molecular dynamics (MD) simulations using the Amber22 package (Salomon-Ferrer et al., 2013) to compare the stability and binding affinity of linear peptides from the test set with cyclic peptides generated by our model. We use the ff14SB force field for proteins and peptides (Maier et al., 2015) with all systems solvated in water, and 150 $nM$ Na$^+$/Cl$^-$ counterions are added to neutralize charges and simulate the normal saline environment (Jorgensen et al., 1983; Li et al., 2024c). The SHAKE algorithm is applied to constrain covalent bonds involving hydrogen atoms (Ryckaert et al., 1977), while non-bonded interactions are truncated at 10.0 Å, with long-range electrostatics treated using the PME method. To estimate peptide binding energies, we further employ MM/PBSA calculations (Genheden & Ryde, 2015). Notably, while MD simulations provide high accuracy in evaluating conformational stability and binding affinity, they are very computationally expensive. Therefore, we randomly select two target proteins from the test set and generate one cyclic peptide using head-to-tail and disulfide bond cyclization strategies for evaluation. More details on the setup of MD are in Appendix C.3.

**Results.** As shown in Fig. 5, the root mean square deviation (RMSD) trajectories of the two linear peptides from the test set exhibit significant fluctuations, indicating vibrate binding conformations. In contrast, the RMSD trajectories of the cyclic peptides generated by our model are quite flat, producing consistently lower RMSD compared to the linear peptides, suggesting that the introduced geometric constraints effectively enhance conformational stability. Table 4 presents the average RMSD values with standard deviations, along with the binding affinity ($\Delta G$) estimated via

*Table 4.* RMSD trajectories from molecular dynamics after 50 ns (average values and standard deviations), along with binding affinities ($\Delta G$) estimated by running simulations with MM/PBSA.

| Peptide | RMSD (Å) | $\Delta G$-MM/PBSA (kcal/mol) |
|---|---|---|
| PDB: 3RC4 | | |
| Linear (test set) | 2.57±0.51 | -9.73 |
| Cyclic (ours) | **1.44±0.23** | **-10.66** |
| PDB: 4J86 | | |
| Linear (test set) | 3.37±0.73 | -15.17 |
| Cyclic (ours) | **1.56±0.40** | **-20.41** |

MM/PBSA simulations. The results indicate that cyclic peptides achieve significantly stronger binding affinities than their linear counterparts, thanks to their enhanced stability in the binding conformations.

### 4.4. Generalization beyond Available Data

In Fig. 6, we visualize the structural embeddings of peptides generated under different cyclization strategies, along with linear peptides from the test set, using ESM2-650M(Lin et al., 2023) and T-SNE (Van der Maaten & Hinton, 2008). The results reveal distinct clusters corresponding to different cyclization strategies, all of which are clearly separated from the linear peptides. This indicates that CP-Composer generalizes well beyond the available data, effectively exploring unseen regions of cyclic peptides.

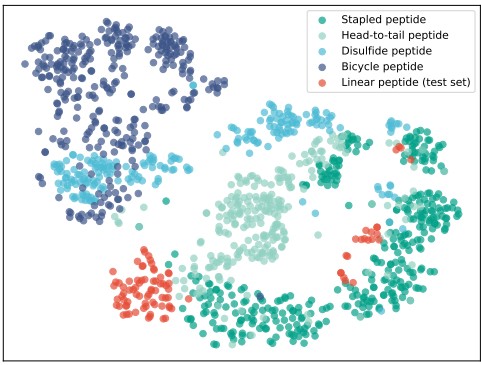

*Figure 6.* T-SNE visualization of ESM embeddings for peptides in the test set and those generated with different cyclization strategies.

## 5. Conclusion

We introduce CP-Composer, a generative framework that enables zero-shot cyclic peptide design via composable geometric constraints. By decomposing complex cyclization patterns into unit constraints, it circumvents the limitation of data, achieves high success rates while preserving fidelity to natural distributions of type and structural statistics, and allows for high-order combinations of cyclization patterns, enabling the design of multi-cycle peptides with customiz-

able strategies. Our framework offers a principled approach to cyclic peptide design, with potential extensions to broader biomolecular applications involving geometric constraints.

## Acknowledgements

This work is jointly supported by the National Key R&D Program of China (No.2022ZD0160502), the National Natural Science Foundation of China (No. 61925601, No. 62376276, No. 62276152), Beijing Nova Program (20230484278), China's Village Science and Technology City Key Technology funding, Beijing Natural Science Foundation (No. QY24249) and Wuxi Research Institute of Applied Technologies.

## Impact Statement

This paper presents work whose goal is to advance the field of Machine Learning. There are many potential societal consequences of our work, none which we feel must be specifically highlighted here.

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

# A. Proofs

## A.1. Proof of Theorem 3.3

For clarity, we restate Theorem 3.3 below.

**Proposition 3.3** (Injective). *Both $f_T$ and $f_D$ are injective. That is, $f(\mathbb{C}^1) = f(\mathbb{C}^2)$ if and only if $\mathbb{C}^1 = \mathbb{C}^2$, where $(f, \mathbb{C}^1, \mathbb{C}^2)$ can be $(f_T, \mathbb{C}^1_T, \mathbb{C}^2_T)$ or $(f_D, \mathbb{C}^1_D, \mathbb{C}^2_D)$. Furthermore, their product function $\tilde{f}(\mathbb{C}_T, \mathbb{C}_D) := (f_T(\mathbb{C}_T), f_D(\mathbb{C}_D))$ is also injective.*

To prove Theorem 3.3, we first prove the following lemma.

**Lemma A.2.** *If $g : \mathbb{R}^J \mapsto \mathbb{R}^K$ is injective, then $f(\mathbb{X}) = \{(i, g(\boldsymbol{k}_i))\}_{i \in \mathcal{V}_{\mathbb{X}}}$ is also injective, where $\mathbb{X} = \{(i, \boldsymbol{k}_i)\}_{i \in \mathcal{V}_{\mathbb{X}}}$.*

*Proof.* $f(\mathbb{X}^1) = f(\mathbb{X}^2) \iff \{(i, g(\boldsymbol{k}_i^1))\}_{i \in \mathcal{V}_{\mathbb{X}^1}} = \{(i, g(\boldsymbol{k}_i^2))\}_{i \in \mathcal{V}_{\mathbb{X}^2}} \iff \mathcal{V}_{\mathbb{X}^1} = \mathcal{V}_{\mathbb{X}^2} := \mathcal{V}_{\mathbb{X}}, g(\boldsymbol{k}_i^1) = g(\boldsymbol{k}_i^2), \forall i \in \mathcal{V}_{\mathbb{X}} \iff \mathcal{V}_{\mathbb{X}^1} = \mathcal{V}_{\mathbb{X}^2} := \mathcal{V}_{\mathbb{X}}, \boldsymbol{k}_i^1 = \boldsymbol{k}_i^2, \forall i \in \mathcal{V}_{\mathbb{X}} \iff \{(i, \boldsymbol{k}_i^1)\}_{i \in \mathcal{V}_{\mathbb{X}^1}} = \{(i, \boldsymbol{k}_i^2)\}_{i \in \mathcal{V}_{\mathbb{X}^2}} \iff \mathbb{X}^1 = \mathbb{X}^2$, where the third deduction step leverages the injectivity of function $g$. $\square$

Now we are ready to prove Theorem 3.3.

*Proof.* We first prove the injectivity of $f_T$. We choose $g$ to be the one-hot encoding function One-hot$(\cdot) : \mathbb{R} \mapsto \mathbb{R}^K$. It is straightforward that this function is injective. By leveraging Lemma A.2, the proof is completed.

For the injectivity of $f_D$, similarly we instantiate $g$ as the RBF feature map $\phi(\cdot) : \mathbb{R} \mapsto \mathbb{R}^\infty$. Such map is injective, since $\|\phi(d_1) - \phi(d_2)\|^2 = <d_1, d_1> + <d_2, d_2> -2 <d_1, d_2> = 1 + 1 - 2\exp(-\gamma\|d_1 - d_2\|^2)$, which implies $\phi(d_1) = \phi(d_2) \iff d_1 = d_2$, hence injectivity. By leveraging Lemma A.2, the proof is completed.[2]

Since both $f_T$ and $f_D$ are injective, $(f_T(\mathbb{C}^1_T), f_D(\mathbb{C}^1_D)) = (f_T(\mathbb{C}^2_T), f_D(\mathbb{C}^2_D)) \iff f_T(\mathbb{C}^1_T) = f_T(\mathbb{C}^2_T), f_D(\mathbb{C}^1_D) = f_D(\mathbb{C}^2_D) \iff \mathbb{C}^1_T = \mathbb{C}^2_T, \mathbb{C}^1_D = \mathbb{C}^2_D \iff (\mathbb{C}^1_T, \mathbb{C}^1_D) = (\mathbb{C}^2_T, \mathbb{C}^2_D)$. Therefore the product function $\tilde{f}(\mathbb{C}_T, \mathbb{C}_D) := (f_T(\mathbb{C}_T), f_D(\mathbb{C}_D))$ is also injective, which concludes the proof. $\square$

## A.2. Equivariance

**Proposition A.3** (Equivariance). *The conditional score $\boldsymbol{\epsilon}_\theta(\mathcal{G}_{\boldsymbol{z}}^{(t)}, \mathbb{C}, t)$ is E(3)-equivariant, where $\mathbb{C}$ is $\mathbb{C}_T$ or $\mathbb{C}_D$.*

The proof is straightforward since our encodings of $\mathbb{C}_T$ and $\mathbb{C}_D$ are both E(3)-invariant, therefore the E(3)-equivariance of the score is preserved, following the proof in Kong et al. (2023).

# B. Decompositions of Cyclic Strategies

As illustrated in Fig. 1, cyclic peptides are looped by four strategies, each of which can be decomposed into unit geometric constraints defined in Sec. 3.2 as follows. Specifically, the pair $(i, l_i)$ indicates a type constraint that node $i$ is required to be type $l_i$, and the triplet $(i, j, d_{ij})$ means a distance constraint that the pairwise distance between node $i, j$ should be $d_{ij}$.

**Stapled peptide.** Given a lysine (K) located at index $i$, a stapled peptide can be formed via a covalent linkage between the lysine and either an aspartic acid (D) at $i + 3$, with constraints as

$$\mathbb{C}_{\text{Stapled-D},i} = (\{(i, K), (i + 3, D)\}, \{(i, i + 3, d_{KD})\}), \tag{11}$$

or a glutamic acid (E) at $i + 4$, with constraints as

$$\mathbb{C}_{\text{Stapled-E},i} = (\{(i, K), (i + 4, E)\}, \{(i, i + 4, d_{KE})\}), \tag{12}$$

where $d_{KD}, d_{KE}$ are the lengths of covalent linkages between the K-D and K-E pairs, respectively.

---

[2]In practice we truncate the infinite-dimensional feature space by setting a limit on the number of bases, similar to Schütt et al. (2018).

**Head-to-tail peptide.** Given a peptide composed of $N$ amino acids indexed by $0, 1, \cdots, N-1$, an additional amide bond is linked between the head and tail amino acid as

$$\mathbb{C}_{\text{Head-to-tail}} = (\{\}, \{(0, N-1, d_P)\}), \tag{13}$$

where $d_P$ is the length of the amide bond.

**Disulfide peptide.** Connecting two non-adjacent cysteines (C) at $i, j$ with a disulfur bond, a disulfide peptide is constrained by

$$\mathbb{C}_{\text{Disulfide},i,j} = (\{(i, C), (j, C)\}, \{(i, j, d_S)\}), \tag{14}$$

where $d_S$ is the length of the disulfur bond.

**Bicycle peptide** To link the three cysteines (C) at $i, j, k$, a bicycle peptide is constrained by

$$\mathbb{C}_{\text{Bicycle},i,j,k} = (\{(i, C), (j, C), (k, C)\}, \{(i, j, d_T), (i, k, d_T), (j, k, d_T)\}), \tag{15}$$

where $d_T$ is the side length of the equilateral triangle formed by the centered 1,3,5-trimethylbenzene.

## C. Implementation Details

### C.1. Energy-based classifier guidance

With the definition of the geometric constraints, we now introduce their corresponding *energy function*, a scalar function that evaluates the satisfaction of the constraint given the input geometric graph.

**Definition C.1** (Energy function of a constraint). An energy function of constraint $\mathbb{C}$ is a differentiable function $g_{\mathbb{C}}(\cdot)$ : $\mathcal{X} \mapsto \mathbb{R}_{\geq 0}$, such that $g_{\mathbb{C}}(\mathcal{G}) = 0$ if $\mathcal{G} \in \mathcal{X}$ satisfies the constraint $\mathbb{C}$ and $g_{\mathbb{C}}(\mathcal{G}) \neq 0$ otherwise.

Intuitively, the energy function serves as an indicator of constraint satisfaction, following the conventional way of handling equality constraints (Bertsekas, 2014).

One naive way to tackle inverse problem is to directly optimize the energy function (Yang et al., 2020; Goldenthal et al., 2007) of the constraint with respect to the initial latents $\mathcal{G}_{\boldsymbol{z}}^{(T)}$, since its minima correspond to the data points $\mathcal{G}$ that satisfy the constraint. However, the large number of sampling steps $T$ required by diffusion models makes the optimization computationally prohibitive, as the gradient needs to be backpropagated through the denoiser $T$ times. Moreover, the energy function is not guaranteed to be convex, which further troubles the optimization.

Energy-based classifier guidance has been introduced to inject constraint as guidance of diffusion sampling in a soft and iterative manner. In our setting, we can pair up $p_t(\mathbb{C}|\mathcal{G}_{\boldsymbol{z}})$ and the energy function through Boltzmann distribution, *i.e.*, $p_t(\mathbb{C}|\mathcal{G}_{\boldsymbol{z}}) = \exp(-g_{\mathbb{C}}(\mathcal{D}_{\xi}(\mathcal{G}_{\boldsymbol{z}})))/Z$, where $Z$ is the normalizing constant. In this way, we have,

$$\nabla_{\mathcal{G}_{\boldsymbol{z}}} \log p_t(\mathcal{G}_{\boldsymbol{z}}|\mathbb{C}) = \nabla_{\mathcal{G}_{\boldsymbol{z}}} \log p_t(\mathcal{G}_{\boldsymbol{z}}) - w \nabla_{\mathcal{G}_{\boldsymbol{z}}} g_{\mathbb{C}}(\mathcal{D}_{\xi}(\mathcal{G}_{\boldsymbol{z}})), \tag{16}$$

where $w \in \mathbb{R}$ is added to control the guidance strength. Performing such sampling procedure is equivalent to sampling from the posterior (Ye et al., 2024):

$$p(\mathcal{G}_{\boldsymbol{z}}|\mathbb{C}) \coloneqq p(\mathcal{G}_{\boldsymbol{z}}) \exp(-w g_{\mathbb{C}}(\mathcal{D}_{\xi}(\mathcal{G}_{\boldsymbol{z}})))/Z, \tag{17}$$

which concentrates the density more on the regions with lower energy function value, biasing the sampling towards data points better satisfying the constraint $\mathbb{C} = (\mathbb{C}_T, \mathbb{C}_D)$.

In our implementation, we adopt the guidance function in Kong et al. (2024) as the energy function $g_{\mathbb{C}}$. In particular, the choice of $w$ significantly influences the generation quality. A larger $w$ typically enhances control strength but degrades generation quality when becoming excessively large. To strike a balance between controllability and quality, we conduct a sweep across various $w$ values and ultimately employ $w \in \{10, 30, 50\}$ for energy-based classifier guidance. The best performance across different $w$ values is reported for all conditions.

## C.2. Distance Constraints as Edge-Level Control

To inject the edge-level control into the model, we apply the adapter mechanism by adding an additional dyMEAN block (Kong et al., 2023) to each layer, and changing the message passing process into

$$\{(\boldsymbol{h}_i^{(l+0.5)}, \vec{\boldsymbol{X}}_i^{(l+0.5)})\}_{i \in \mathcal{V}} = \text{AME}(\{(\boldsymbol{h}_i^{(l)}, \vec{\boldsymbol{X}}_i^{(l)})\}_{i \in \mathcal{V}}, \{\boldsymbol{d}_{ij}\}_{(i,j) \in \mathcal{E}_D}, \mathcal{E}_D), \tag{18}$$

$$\{(\boldsymbol{h}_i^{(l+1)}, \vec{\boldsymbol{X}}_i^{(l+1)})\}_{i \in \mathcal{V}} = \text{AME}(\{(\boldsymbol{h}_i^{(l+0.5)}, \vec{\boldsymbol{X}}_i^{(l+0.5)})\}_{i \in \mathcal{V}}, \varnothing, \mathcal{E}), \tag{19}$$

where $\mathcal{E}_D \subseteq \mathcal{E}$ is the set of constrained edges, and AME is the Adaptive Multi-Channel Equivariant layer proposed in Kong et al. (2023). Readers are referred to the original paper for further details.

## C.3. Molecular Dynamics

We perform molecular dynamics (MD) simulations to assess the stability and binding affinity of linear peptides from the test set and cyclic peptides generated by our model. Simulations are conducted using the Amber22 package with the CUDA implementation of particle-mesh Ewald (PME) MD and executed on GeForce RTX 4090 GPUs (Salomon-Ferrer et al., 2013). For system preparation, the ff14SB force field is applied to proteins and peptides (Maier et al., 2015). All systems are solvated to a 10 Å truncated octahedron transferable intermolecular potential three-point (TIP3P) water box and 150 $nM$ Na$^+$/Cl$^-$ counterions are added to neutralize charges and simulate the normal saline environment (Jorgensen et al., 1983; Li et al., 2024c). Prior to equilibration, two rounds of energy minimization are performed: the first relaxes solvent molecules and Na$^+$/Cl$^-$ counterions while keeping all other atoms fixed, and the second relaxes all atoms without constraints. The systems are then gradually heated from 0 K to 310 K over 500 ps under harmonic restraints of 10 kcal $\cdot$ mol$^{-1} \cdot$ Å$^{-2}$ on proteins and peptides. Subsequently, equilibration is carried out at 300 K and 1 bar under NPT conditions, with harmonic restraints on protein and ligand atoms progressively reduced from 5.0 to 3.0, 1.0, 0.5, and finally 0.1 kcal $\cdot$ mol$^{-1} \cdot$ Å$^{-2}$ spanning a total of 2.5 ns. Production simulations are performed with temperature (300 K) and pressure (1 bar) using the Langevin thermostat and Berendsen barostat, respectively. The SHAKE algorithm is applied to constrain covalent bonds involving hydrogen atoms (Ryckaert et al., 1977), while non-bonded interactions are truncated at 10.0 Å, with long-range electrostatics treated using the PME method. To estimate peptide binding energies, we further employ MM/PBSA calculations (Genheden & Ryde, 2015). While MD simulations provide high accuracy in evaluating conformational stability and binding affinity, they are computationally expensive. Therefore, we randomly select two target proteins from the test set and generate one cyclic peptide using head-to-tail and disulfide bond cyclization strategies for evaluation.

## C.4. Hyperparamter details

We train CP-Composer on a 24G memory RTX 3090 GPU with AdamW optimizer. For the autoencoder, we train for up to 100 epochs and save the top 10 models based on validation performance. We ensure that the total number of edges (scaling with the square of the number of nodes) does not exceed 60,000. The initial learning rate is set to $10^{-4}$ and is reduced by a factor of 0.8 if the validation loss does not improve for 5 consecutive epochs. Regarding the diffusion model, we train for no more than 1000 epochs. The learning rate is $10^{-4}$ and decay by 0.6 and early stop the training process if the validation loss does not decrease for 10 epochs. During the training process, we set the guidance strength as 1 for sampling at the validation stage. The structure details of the autoencoder and the diffusion model are the same as Kong et al. (2024). For the RBF kernel, we use 32 feature channels.

# D. Further Analysis

## D.1. Necessity of RBFs

We evaluate the influence of the RBFs to the quality of the generation of peptide under most difficult setting: Bicycle peptide (26 samples in test set). In Table 5, Based on the validation and parameter sensitivity study, we can conclude the necessity of RBF design to support the distance control. Further, an saturation beyond 16 channels is observed, indicating that finite RBFs is enough for empirical performance.

## D.2. Generation efficiency

In Table 6, we show the runtime comparison between our method and the DiffPepBuilder when they both use a 24GB RTX3090 GPU.

*Table 5.* Success rates among different number of RBFs

| Succ.(w=2) | Bicycle peptide |
| --- | --- |
| RBFs=0 | 26.92% |
| RBFs=16 | 30.76% |
| RBFs=32 | 30.76% |

*Table 6.* Runtime of our method and DiffPepBuilder

|  | CP-Composer | DiffPepBuilder |
| --- | --- | --- |
| second per peptide | 1.42s | 29.94s |

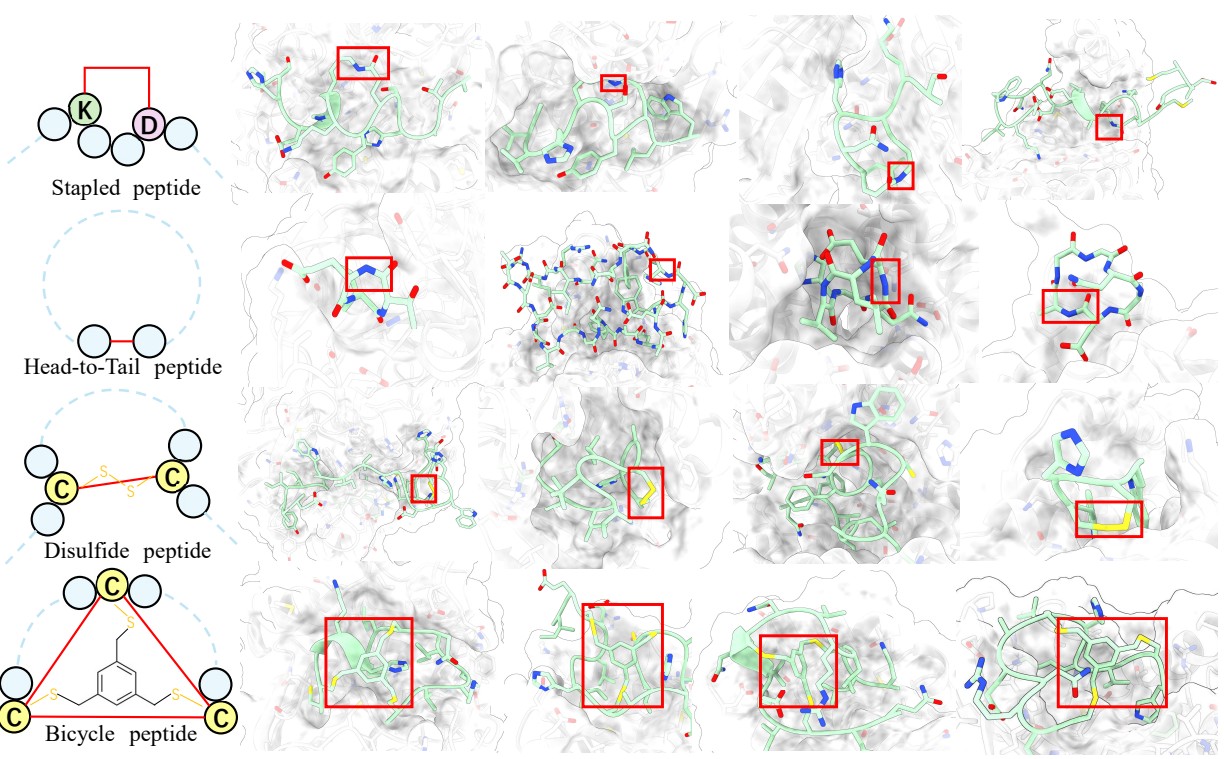

*Figure 7.* Four types of generated cyclic peptides, with the red boxes highlighting the position for cyclization.

# E. Additional Visualizations

In Fig. 7, we show more cases of the stapled, Head-to-tail, disulfur and bicycle peptide.

# F. Code Availability

The codes for our CP-Composer is provided at the link https://github.com/jdp22/CP-Composer_final.

