# OpenReview forum: "Zero-Shot Cyclic Peptide Design via Composable Geometric Constraints"
_ICML.cc/2025/Conference — ICML 2025 poster_

### Official Review · Reviewer_EnLZ · 2025-02-19

**Overall Recommendation:** 3

**Summary:**

This paper presents CP-Composer, a zero-shot cyclic peptide design framework using composable geometric constraints. The key innovation lies in decomposing complex cyclization strategies into type constraints and distance constraints, integrated into a geometric graph diffusion model via conditional encoding. Experiments demonstrate superior success rates across four cyclization tasks compared to baselines, while molecular dynamics (MD) simulations confirm enhanced conformational stability and binding affinity over linear peptides. The framework supports flexible cyclic peptide designs without requiring training data, offering a promising tool for customizable drug discovery.

## update after rebuttal

Thanks for your response, I will keep my positive score.

**Claims And Evidence:**

These claims are supported by convincing evidence, both theoretical and empirically validated. However, there are some small holes in the validation (robustness of MD and ablation studies) that slightly weaken the argument.

**Essential References Not Discussed:**

No essential references have been omitted.

**Experimental Designs Or Analyses:**

The experimental design adequately addresses core claims but lacks rigor in validation: small MD sample size and missing ablation studies.

**Methods And Evaluation Criteria:**

The proposed methods and evaluation criteria are largely appropriate and well-aligned with the problem of cyclic peptide design, but certain aspects could be strengthened.
- Limited MD Validation: Only two test cases are simulated. A larger sample size and error analysis (e.g., standard deviations across replicates) would improve confidence.
- The paper does not clarify whether test-time cyclization strategies (e.g., disulfide bonds) involve constraints entirely absent from training. If training data includes residues like cysteines (common in disulfide bonds), the "zero-shot" claim might overstate novelty.

**Other Comments Or Suggestions:**

Add runtime metrics (e.g., seconds per peptide generation) to Appendix C.4 for scalability assessment.

**Other Strengths And Weaknesses:**

## Weaknesses:
- Method: lack of ablation studies.
- Validation: Small MD Sample Size: Only two test cases are simulated, reducing confidence in stability claims. Diversity metrics are missing.
- Synthesizability: No discussion of whether generated peptides are chemically feasible or compatible with synthesis pipelines.

**Questions For Authors:**

Were generated peptides assessed for chemical synthesizability (e.g., using SAscore or RAscore)? If not, how might impractical structures affect real-world utility?

**Relation To Broader Scientific Literature:**

CP-Composer addresses the unique challenges of cyclic peptide design through composable constraints and zero-point learning, providing a new paradigm for customizable molecule generation with broad scientific and industrial relevance,

**Theoretical Claims:**

The theoretical claims are mathematically correct under ideal assumptions, but practical implementations introduce approximations (finite RBFs) that weaken injectivity guarantees. The reliance on external proofs for equivariance is acceptable but leaves a minor risk of inherited errors. Overall, the proofs are valid but lack robustness analysis for real-world settings.

---

> ### Author Rebuttal · Authors · 2025-03-31
>
> > Q1: Limited MD Validation: Only two test cases are simulated. A larger sample size and error analysis (e.g., standard deviations across replicates) would improve confidence.
>
> Sorry for the limitation, due to the time and resource limitation, we adopted the rhotheta score as an auxiliary metric to validate our generated peptides. Specifically, we first relax the peptide using a dedicated force field to establish a stable cyclic conformation; thereafter, we further optimize the structure with a rhotheta relaxer and compute the corresponding rhotheta score. Our findings indicate that 85.54% of the target peptides achieve a negative rhotheta score, signifying an energetically favorable state. The rhotheta score is a widely used metric for peptide evaluation, providing an efficient method for assessing large sample sizes.
>
> > Q2: The paper does not clarify whether test-time cyclization strategies (e.g., disulfide bonds) involve constraints entirely absent from training. If training data includes residues like cysteines (common in disulfide bonds), the "zero-shot" claim might overstate novelty.
>
> Sorry for the confusion. Here "zero-shot" means that the specific combination of constraints required for cyclic peptides is not present during training. While the training data includes individual unit constraints (e.g., cysteine residues), the exact combination needed for cyclization is not observed. Our method learns combinations of constraints in linear peptides during training and generalizes to their combinations during inference to generate cyclic peptides.
>
> > Q3: The theoretical claims are mathematically correct under ideal assumptions, but practical implementations introduce approximations (finite RBFs) that weaken injectivity guarantees. The reliance on external proofs for equivariance is acceptable but leaves a minor risk of inherited errors. Overall, the proofs are valid but lack robustness analysis for real-world settings... The experimental design adequately addresses core claims but lacks rigor in validation: ... missing ablation studies.
>
> We evaluate the influence of the RBFs to the quality of the generation of peptide under most difficult setting: Bicycle peptide(26 samples in test set):
>
> |  Succ.(w=2) | Bicycle peptide |
> |-------|-----------:|
> | RBFs=0  |26.92%   |
> | RBFs=16 | 30.76% |
> | RBFs=32 | 30.76% |
>
> Based on the validation and parameter sensitivity study, we can conclude the necessity of RBF design to support the distance control. Further, an saturation beyond 16 channels is observed, indicating that finite RBFs is enough for empirical performance.
>
> > Q4: Reproducibility: While code is provided, computational resource requirements are underspecified.
>
> We train CP-Composer on a 24G memory RTX 3090 GPU with AdamW optimizer. More details can be found in Section C.4.
>
> > Q5: Add runtime metrics (e.g., seconds per peptide generation) to Appendix C.4 for scalability assessment.
>
> Here we show the runtime comparison between our method and the baseline method when they both use a 24GB RTX3090 GPU.
>
> |       | CP-Composer | DiffPepBuilder |
> |-------|-----------:|-----------:|
> | second per peptide | 1.42s     | 29.94s     |
>
> > Q6: Were generated peptides assessed for chemical synthesizability (e.g., using SAscore or RAscore)? If not, how might impractical structures affect real-world utility?
>
> The SA score is primarily designed for assessing the synthesizability of small molecules and is not directly applicable to peptides, which follow a distinct synthesis methodology. For peptides, higher hydrophobicity can lead to aggregation and hinder synthesis. Thus, we analyze the distribution of GRAVY scores [1], which measure peptide hydrophobicity. The results, provided in the linked analysis（https://anonymous.4open.science/r/Rebuttal-68EF/readme.md), demonstrate that the generated peptides exhibit a hydrophobicity distribution similar to that of natural peptides, suggesting that they are likely to be synthesizable.
>
> Reference：
> [1] Kyte, Jack, and Russell F. Doolittle. "A simple method for displaying the hydropathic character of a protein." Journal of molecular biology 157.1 (1982): 105-132.

---

### Official Review · Reviewer_ctzr · 2025-03-14

**Overall Recommendation:** 2

**Summary:**

Cyclic peptides exhibit superior biochemical properties and can be used to address emerging medical needs. However, due to the limited availability of training data, research on cyclic peptide design remains scarce. This paper introduces a novel generative model that employs a composable constraint approach, enabling the generation of cyclic peptides during inference, even without cyclic peptide training data. Specifically, the proposed method decomposes complex cyclization patterns into unit constraints, which are incorporated into a diffusion model through geometric conditioning on nodes and edges. During training, the model learns unit constraints and their various combinations from linear peptides. At inference, specific cyclization constraint combinations are imposed as input. Experimental results demonstrate that, despite being trained solely on linear peptides, the model can generate diverse target-binding cyclic peptides, achieving a significant improvement in success rate.

**Claims And Evidence:**

Yes, the proposed claims are clear and well-supported by corresponding evidence.

**Essential References Not Discussed:**

No

**Experimental Designs Or Analyses:**

Yes, I have checked the experimental design and results of the paper and identified the following issues.

### Design of the experiments

1. The paper only compares CP-Composer with **PepGLAD** and the **EG method**, without including other diffusion- or flow-based methods. I believe the authors should compare their approach with **basic diffusion models** and some **recent conditional generative models**, rather than only comparing it to a single backbone model.
2. For the generated molecules, the authors only evaluate whether the cyclic peptide is successfully generated but do not assess the validity of the generated molecules. In addition to the success rate, they should also consider whether there are any unreasonable atomic types or coordinates in the generated structures, which could be measured using an appropriate metric.

### Results

In Table 2, the success rate of **“2\*Stapled”** does not change with variations in **w**. While the paper acknowledges this phenomenon, it does not discuss the possible reasons behind it. What could be causing this? Did the authors check whether the unsuccessful cases at **w = 2, 2.5, and 3** were the same set of samples?

### Writing

The paper does not provide the deetailed explanation for the metrics. For the **AA-KL** and **B-KL** metrics, it does not clearly explain how the distribution divergence is computed.

**Methods And Evaluation Criteria:**

Yes, the proposed method in this paper makes sense for peptide design problems, particularly for cyclic peptide generation.

Regarding the method details, the paper proposes using a composable constraint approach to enforce cyclic peptide constraints. Specifically, it combines **type constraints** and ** distance constraints** as the generation constraints for cyclic peptides. This approach is quite novel and makes sense. However, these constraints may not be entirely **independent** but rather **intertwined**, which the authors have not discussed.

**Other Comments Or Suggestions:**

The writing can be improved. I notice several typos:

* C.4. Hyperparamter details: Hyperparameter
* The overal workflow is depicted in Fig. 2. --> overall
* The sampled is acquired (near eq. 8): --> sample

I’ve noticed these errors, and there may be others as well. The author should carefully proofread the text to ensure accuracy.

**Other Strengths And Weaknesses:**

### Pros

* The article proposes a compositional conditional generative model that can generate target peptides in a zero-shot manner by combining multiple constraints.
* The article provides a proof of the method’s validity.

### Cons

* The experimental section is not sufficiently comprehensive; more baselines need to be compared.
* Additionally, some experimental phenomena require more detailed discussion (see the experimental section for details).
* There are some grammatical issues and typos, and the writing needs improvement.

**Questions For Authors:**

I think this method is simple and novel, and the compositionality of the conditions has been proven to be injective. However, in real-world scenarios, is the combination of these conditions truly linear? (Is adding condition A and condition B really equivalent to A + B?) What do you think about this? Are there any methods to explore this further?

**Relation To Broader Scientific Literature:**

The proposed composable constraint approach for generating cyclic peptides with limited data is novel and can also provide insights for generation tasks in other data-scarce domains.

**Theoretical Claims:**

Yes, I have checked the experimental design and results of the paper and identified the following issues.

---

> ### Author Rebuttal · Authors · 2025-03-31
>
> Thanks for the insightful comments which help improve the quality of our paper!
>
> > Q1: The type and distance constraints may not be independent but rather intertwined, which the authors have not discussed.
>
> Sorry for the confusion. We ensure that the constraints are jointly feasible by considering their dependencies both in the data and the model. First, the constraint combinations are sampled from the joint distribution derived from real-world data, based on the dataset during training and based on chemical knowledge during inference. Second, our model inherently accounts for these dependencies, as the constraints are input together and fused within the hidden layers to output the conditional denoising score.
>
> > Q2: I believe the authors should compare their approach with basic diffusion models and some recent conditional generative models.
>
> Thanks for the suggestion. We further include two more baseline methods, CADS, an advanced diffusion conditional sampler, and DiffPepBuilder, a model specifically designed for disulfide peptides, as below:
>
> |  Succ.     | Staple | Head-Tail | Disulfide | Bicycle |
> |-------|-----------:|-----------:|-----------:|-----------:|
> |ours|21.42%|65.11%|41.25%|30.76%
> | CADS | 27.14%     | 45.54%     | 3.75%      | 3.85%      |
> | DiffPepBuilder| -     | -     | 23.07%      | -      |
>
> For CADS sampler, we set the w=2 and use the sampler with initial noise scale as 0.25. The results indicate that CADS is acceptable under simple constraints but struggles with more complex constraints like disulfide and bicyclic peptides. Notably, despite being a general zero-shot model, our approach outperforms DiffPepBuilder, which is specifically designed for disulfide peptides.
>
>
> > Q3: For the generated molecules, the authors only evaluate whether the cyclic peptide is successfully generated but do not assess the validity of the generated molecules. In addition to the success rate, they should also consider whether there are any unreasonable atomic types or coordinates in the generated structures, which could be measured using an appropriate metric. For the AA-KL and B-KL metrics, it does not clearly explain how the distribution divergence is computed.
>
> Sorry for the confusion. We assess the validity of generated peptides by evaluating their residue composition and structural features to ensure they resemble natural peptides. Specifically, we use KL divergence on residue types and dihedral angles. AA-KL measures the KL divergence between amino acid distributions in generated peptides and reference peptides, excluding the controlled amino acid types. B-KL quantifies the KL divergence between dihedral angle distributions of generated and reference peptides, ensuring realistic backbone conformations. Additionally, we assess the structural stability of generated peptides using Rosetta energy scores. Our results show that 85.54% of cyclic peptides satisfying constraints achieve negative Rosetta scores, indicating their physical plausibility. This suggests that the model generates valid and stable cyclic peptides.
>
>
> > Q4: In Table 2, the success rate of “2*Stapled” does not change with variations in w. While the paper acknowledges this phenomenon, it does not discuss the possible reasons behind it. What could be causing this? Did the authors check whether the unsuccessful cases at w = 2, 2.5, and 3 were the same set of samples?
>
> We appreciate the insightful question! Upon examining the failure cases for w=2,2.5, and 3, we found that they are nearly identical. This suggests that, beyond a certain threshold, these cases have already collapsed into failure modes. As a result, further increasing the constraint strength does not impact the success rate.
>
>
> > Q5: However, in real-world scenarios, is the combination of these conditions truly linear? (Is adding condition A and condition B really equivalent to A + B?) What do you think about this? Are there any methods to explore this further?
>
> Thank you for your thoughtful question. We think there are some combinations not directly addictive. For example, a bicycle peptide may struggle to also satisfy the stapled peptide constraints, which we have observed in zero success rates for such cases. In most application scenarios, meaningful combinations are ensured by expertise in structural chemistry and biology. Additionally, low success rates for certain combinations can help identify infeasible or unrealistic configurations.

---

### Official Review · Reviewer_1riF · 2025-03-14

**Overall Recommendation:** 4

**Summary:**

The paper proposes CP-Composer, a novel diffusion-based generative framework for zero-shot cyclic peptide design. The authors motivate their approach by highlighting the data scarcity problem in cyclic peptide design, where obtaining experimental data for diverse cyclization patterns is challenging. The key innovation presented in the paper is the decomposition of complex cyclisation constraints into simpler, composable "unit constraints" representing type and distance relationships. This decomposition allows the model to be trained on more readily available linear peptide data, learning these unit constraints and their combinations, and then generalizing to unseen, complex cyclic constraints at inference time. The method is evaluated on several cyclisation strategies (stapled, head-to-tail, disulfide, and bicycle peptides), demonstrating improved success rates in generating valid cyclic structures compared to existing baselines. The authors also present results from molecular dynamics simulations, providing evidence for the stability and binding affinity of the generated peptides. The primary application is the design of novel cyclic peptides, which have potential as therapeutic drug candidates. Overall, the paper presents a well-written and technically sound approach, contributing to the fields of peptide design and constrained generative modelling.

**Claims And Evidence:**

The central claim of the paper is that CP-Composer enables effective zero-shot cyclic peptide design via composable unit constraints. However, to more convincingly demonstrate the advantages of the proposed approach and solidify its claims, the authors should expand the baseline comparison and provide a more rigorous theoretical justification for the guidance mechanism. Specifically, the comparison should be extended to include more advanced guidance techniques, such as the unified guidance framework presented in [1], which could readily incorporate the proposed unit constraints, a simply classifier-free guidance together with the CADS sampler [2]. Additionally, comparing against any existing, even if limited, methods specifically designed for cyclic peptide generation is important for contextualizing the performance gains. Moreover, the connection between the ideal conditional score (Eq.2) and the implemented guided score (Eq.3) should be strengthened. A more formal derivation, potentially drawing inspiration from the diffusion posterior sampling approach of [3] or the generalized h-transform [4] (who essentially end up with the same terms as Eq.3 for controlled generation), would provide a more solid theoretical foundation for the chosen guidance strategy. Addressing these points would enhance the paper's claims.


&nbsp;

[1] Ayadi, S., Hetzel, L., Sommer, J., Theis, F., & Günnemann, S. (2024). Unified Guidance for Geometry-Conditioned Molecular Generation. Advances in Neural Information Processing Systems, 37, 138891-138924.

[2] Sadat, S., Buhmann, J., Bradely, D., Hilliges, O., and Weber, R. M. Cads: Unleashing the diversity of diffusion models through condition-annealed sampling. arXiv preprint arXiv:2310.17347, 2023.

[3] Chung, H., Kim, J., Mccann, M. T., Klasky, M. L., & Ye, J. C. (2022). Diffusion posterior sampling for general noisy inverse problems. arXiv preprint arXiv:2209.14687.

[4] Denker, A., Vargas, F., Padhy, S., Didi, K., Mathis, S., Barbano, R., ... & Lio, P. (2024). DEFT: Efficient Fine-tuning of Diffusion Models by Learning the Generalised $ h $-transform. Advances in Neural Information Processing Systems, 37, 19636-19682.

**Essential References Not Discussed:**

No essential works appear to be missing.

**Experimental Designs Or Analyses:**

I reviewed the experimental designs, as detailed above. The primary concerns are the limited choice of baselines, which hinders a comprehensive evaluation of performance, and the need for a more rigorous theoretical justification connecting the proposed guidance mechanism to established works in the field.

**Methods And Evaluation Criteria:**

The proposed methods and evaluation criteria are generally appropriate for the problem of zero-shot cyclic peptide design, utilising relevant metrics. However, the experimental design could be strengthened to better support the claims and contextualise CP-Composer's performance.

- Section 4.1 (Zero-Shot Cyclic Peptide Generation): This section is meant to demonstrate the core capability of generating different cyclic peptide types in a zero-shot setting. However, the limited baselines (no existing cyclic peptide methods, basic energy guidance) weaken the comparison. Moreover, a more thorough analysis of failure cases would be insightful to understand the method's limitations.
- Section 4.2 (Flexibility in High-Order Combinations): Here, the authors show the framework's ability to handle complex, combined constraints, supporting the claim of generalisability. However, the relative advantage of CP-Composer is difficult to assess and the section would benefit from a more extensive comparisons with advanced baselines, which should also generalise to this more challenging setting.
- Section 4.3 (Evaluations by Molecular Dynamics): This section strengthens claims of practical utility with MD simulations, but the presented scope (two targets, two cyclization strategies) restricts general conclusions. More simulations, potentially build on faster approximate methods, would be great.
- Section 4.4 (Generalisation beyond Available Data): Here, the authors use t-SNE plots to visualise the generation of non-linear peptides. This could be strengthened by adding quantitative sequence analysis and a comparison of training and generated peptide sequences.

**Other Comments Or Suggestions:**

No.

**Other Strengths And Weaknesses:**

**Originality**: The setting of zero-shot cyclic peptide design via composable unit constraints, enabling training on readily available linear peptide data, is a novel contribution. However, the derivation of the guidance mechanism lacks novelty and could benefit from a more rigorous theoretical grounding, as discussed in previous sections. This makes the decomposition of complex cyclisation constraints into simpler, composable units (type and distance) the primary source of originality.

**Significance**: The work addresses a significant challenge in drug discovery, and if the claims were fully supported by a more comprehensive experimental evaluation, the work would be relevant for the peptide design community.

**Clarity**: The paper is well-written and structured logically, with core concepts and the proposed method clearly explained. The figures are informative, enhancing overall clarity.

**Questions For Authors:**

1) Could you elaborate on the specific challenges posed by Disulfide and Bicycle peptides for the baselines? Here, CP-Composer achieves the best success rates (even better than Stapled and HT) while the baselines fail.
2) Have you investigated any techniques to further improve the success rates, particularly for the higher order combinations?

&nbsp;

____
I am willing to increase my score if the authors address the raised concerns.

**Relation To Broader Scientific Literature:**

The paper cites relevant works in the areas of diffusion models, geometric deep learning, and peptide design. However, the discussion of related work and the theoretical grounding could be strengthened. The discussion of diffusion guidance could be significantly improved by referencing and incorporating more advanced techniques, such as the unified guidance framework in [1] or the condition-annealed sampling approach in [2]. Furthermore, a more rigorous theoretical justification connecting the proposed guidance to established work like DPS [2] and DEFT [3] is needed. This would better position the work within the existing literature and address the limitations in the baseline comparisons.

**Theoretical Claims:**

I did not check the proofs in detail, focusing instead on the overall conceptual
soundness and experimental validation.

---

> ### Author Rebuttal · Authors · 2025-03-31
>
> Thank you for your suggestions!
>
> > Q1:Comparing against any existing methods specifically designed for cyclic peptide generation is important for contextualizing the performance gains.
>
> We include CADS, an advanced diffusion conditional sampler, and DiffPepBuilder, a model specifically designed for disulfide peptides, as below:
>
> |Succ.|Staple|Head-Tail|Disulfide|Bicycle|
> |-|-:|-:|-:|-:|
> |Ours|21.42%|65.11%|41.25%|30.76%
> |CADS|27.14%|45.54%|3.75%|3.85%|
> |DiffPepBuilder|-|-|23.07%|-|
>
> For CADS sampler, we set the w=2 and use the initial noise scale as 0.25. CADS is acceptable under simple constraints but struggles with more complex constraints like disulfide and bicyclic peptides. Notably, our zero-shot approach outperforms DiffPepBuilder, which is specifically designed for disulfide peptides.
>
> > Q2: The connection between the ideal conditional score (Eq.2) and the implemented guided score (Eq.3) should be strengthened.
>
> In fact, the rationale of Eq.2 and Eq.3 are linked by the following distribution
> $$ \tilde{p}\_t(\mathcal{G}\_\mathbf{z}^{(t)}|\mathbb{C}) \propto p\_t(\mathcal{G}\_\mathbf{z}^{(t)})  p\_t(\mathbb{C}|\mathcal{G}\_\mathbf{z}^{(t)})^w, $$
> with the corresponding conditional score
> $$\nabla\_{\mathcal{G}\_\mathbf{z}^{(t)}}\log \tilde{p}\_t(\mathcal{G}\_\mathbf{z}^{(t)}|\mathbb{C}) =\nabla\_{\mathcal{G}\_\mathbf{z}^{(t)}}\log p\_t(\mathcal{G}\_\mathbf{z}^{(t)})  +w\nabla\_{\mathcal{G}\_\mathbf{z}^{(t)}}\log p\_t(\mathbb{C}|\mathcal{G}\_\mathbf{z}^{(t)})\approx\epsilon\_\theta(\mathcal{G}\_\mathbf{z}^{(t)},t) + w\nabla\_{\mathcal{G}\_\mathbf{z}^{(t)}}\log p\_t(\mathbb{C}|\mathcal{G}\_\mathbf{z}^{(t)})  .$$
>
> In particular, Eq.2 directly models  $\log p\_t(\mathbb{C}|\mathcal{G}\_\mathbf{z}^{(t)})$ by an externally trained energy function. Eq.3 instead follows classifier-free guidance by rewriting $\nabla\_{\mathcal{G}\_\mathbf{z}^{(t)}}\log p\_t(\mathbb{C}|\mathcal{G}\_\mathbf{z}^{(t)})=\nabla\_{\mathcal{G}\_\mathbf{z}^{(t)}}\log p\_t(\mathcal{G}\_\mathbf{z}^{(t)}|\mathbb{C})-\nabla\_{\mathcal{G}\_\mathbf{z}^{(t)}}\log p\_t(\mathcal{G}\_\mathbf{z}^{(t)})\approx\epsilon\_\theta(\mathcal{G}\_\mathbf{z}^{(t)},\mathbb{C},t)-\epsilon\_\theta(\mathcal{G}\_\mathbf{z}^{(t)},t),$ which gives Eq.3 after simplification.
>
> > Q3：A more thorough analysis of failure cases would be insightful to understand the method's limitations.
>
> As for failure cases, they commonly occur when excessively large controling strength w degrade molecular quality. We analyzed peptide structures generated with w=6 (https://anonymous.4open.science/r/Rebuttal-68EF/readme.md) and found that they often exhibited strained backbones and unrealistic conformations. This highlights the trade-off between guidance strength and structural integrity, suggesting that an adequate selection strategy for w could further improve robustness.
>
> > Q4: The relative advantage of CP-Composer is difficult to assess and the section would benefit from a more extensive comparisons with advanced baselines, which should also generalise to this more challenging setting.
>
> First, our model allows arbitrary high-order combinations, as shown in the paper. Second, the comparison with DiffPepBuilder below indicate that, despite being a general zero-shot model, our approach outperforms the baseline specifically designed for disulfide peptides.
>
> |2*-S-S-|CP-Composer|DiffPepBuilder|
> |-|-:|-:|
> |Succ.|62.0%| 30.48%|
>
> > Q5: The authors use t-SNE plots to visualise the generation of non-linear peptides. This could be strengthened by adding quantitative sequence analysis and a comparison of training and generated peptide sequences.
>
> Sorry for the confusion. Our current cluster analysis in Figure 6 is already based on the embedding generated from ESM2 model, which outputs embeddings based on the sequence information.
>
> > Q6: Could you elaborate on the specific challenges posed by Disulfide and Bicycle peptides for the baselines?
>
> Disulfide and Bicycle peptides need requires more strict constraints than Stapled peptide and Head-to-tail peptide. Dislfide peptide need one distance constraint unit and two type constraint unit. Bicycle peptides need three distance constraint units and three type constraint units. While the Head-to-tail peptide only needs one distance constraint unit and the Stapled peptide needs one distance constraint and two type constrinis easier to reach as both K-D and K-E are accepted.
>
> > Q7: Have you investigated any techniques to further improve the success rates, particularly for the higher order combinations?
>
> As an initial attempt at zero-shot cyclic peptide generation, our focus is to demonstrate the feasibility and generalizability of our framework, which already shows promising results. However, in more complex constraint combinations, conflicts may arise, potentially driving the generated peptides in divergent directions. Addressing these conflicts to further improve success rates is an important challenge that we leave for future research.

---

> > ### Comment · Reviewer_1riF · 2025-04-05
> >
> > I would like to thank the authors for their efforts during the rebuttal period. I have carefully reviewed their responses, as well as the other reviewers’ comments. For the revised version of the paper, I find it important that the authors include their insights from Q1 and Q2 and update their method discussion based on my original feedback (as outlined in the "Claims and Evidence" section).
> >
> > Assuming the authors will adequately address these points in the final version—particularly by softening the claim in L193–194 and clearly situating their work in relation to the prior literature I referenced—I am raising my score.

---

> > > ### Author Response · Authors · 2025-04-05
> > >
> > > Thank you for your positive feedback and support! We’re glad to hear that our responses have addressed your concerns. Following your valuable suggestions, we will incorporate all rebuttal content into the final version to further enhance the quality of our paper.

---

### Decision · Program_Chairs · 2025-05-01

**Decision:**

Accept (poster)

**Comment:**

The paper introduces CP-Composer, a novel framework for zero-shot cyclic peptide design using composable geometric constraints. The work addresses the challenge of designing cyclic peptides with minimal training data by leveraging a diffusion model and geometric conditioning. The paper is well-received, with notable strengths in its innovative approach, solid experimental results, and clarity in presentation.

Reviewer scores initially varied with scores of 2, 3, and 3 (only 3 reviews were received), reflecting a range of opinions primarily concerning baseline comparisons and theoretical grounding. Post-rebuttal discussions involved authors addressing all major concerns, such as conducting extensive experiments against additional baselines like CADS and DiffPepBuilder, and explaining the connection between the guided and conditional scores more clearly. These clarifications were well-received by reviewers who engaged with the discussion, with one reviewer improving their score from 3 to 4. Note that the reviewer who scored 2 did not respond to the rebuttal despite repeated prompting.

The work's strength lies in its innovative design and potential application to drug discovery. However, the reviewers highlighted that additional baseline comparisons and further theoretical backing for guidance mechanisms would strengthen the claims. The rebuttal addressed these points effectively, reinforcing the paper's validity. The authors also did a good job of acknowledging the limitations of the approach and identifying avenues for future work.

Overall, the paper presents valuable contributions to the field of peptide design and constrained generative modelling, and I recommend accepting it as a solid addition to ICML.